# Clinical Features and Management of Skull Base Fractures in the Pediatric Population: A Systematic Review

**DOI:** 10.3390/children11050564

**Published:** 2024-05-08

**Authors:** Geena Jung, Jorden Xavier, Hailey Reisert, Matthew Goynatsky, Margaret Keymakh, Emery Buckner-Wolfson, Timothy Kim, Ryan Fatemi, Seyed Ahmad Naseri Alavi, Andres Pasuizaca, Pushti Shah, Genesis Liriano, Andrew J. Kobets

**Affiliations:** 1Montefiore Medical Center, Albert Einstein College of Medicine, 111 E 210th Street, Bronx, NY 10461, USA; jorden.xavier@einsteinmed.edu (J.X.); hailey.reisert@einsteinmed.edu (H.R.); margaret.keymakh@einsteinmed.edu (M.K.); emery.bucknerwolfson@einsteinmed.edu (E.B.-W.); timothy.kim@einsteinmed.edu (T.K.); ryan.fatemi@einsteinmed.edu (R.F.); andres.pasuizaca@einsteinmed.edu (A.P.); pushti.shah@einsteinmed.edu (P.S.); 2School of Medicine, Trinity College Dublin, Dublin 2, Ireland; goynatsm@tcd.ie; 3Department of Neurosurgery, Montefiore Medical Center, Bronx, NY 10461, USA; seyedahmad.naserialavi@einsteinmed.edu (S.A.N.A.); gliriano@montefiore.org (G.L.); akobets@montefiore.org (A.J.K.)

**Keywords:** skull base fracture, basilar skull fracture, pediatric, intracranial injury

## Abstract

Pediatric basilar skull fractures (BSFs) are a rare type of traumatic head injury that can cause debilitating complications without prompt treatment. Here, we sought to review the literature and characterize the clinical features, management, and outcomes of pediatric BSFs. We identified 21 relevant studies, excluding reviews, meta-analyses, and non-English articles. The incidence of pediatric BSFs ranged from 0.0001% to 7.3%, with falls from multi-level heights and traffic accidents being the primary causes (9/21). The median presentation age ranged from 3.2 to 12.8 years, and the mean age of patients across all studies was 8.68 years. Up to 55% of pediatric BSFs presented with intracranial hematoma/hemorrhage, along with pneumocephalus and edema. Cranial nerve palsies were a common complication (9/21), with the facial nerve injured most frequently (7/21). While delayed cranial nerve palsy was reported in a few studies (4/21), most resolved within three months post-admission. Other complications included CSF leaks (10/21) and meningitis (4/21). Management included IV fluids, antiemetics, and surgery (8/21) to treat the fracture directly, address a CSF leak, or achieve cranial nerve compression. Despite their rarity, pediatric skull base fractures are associated with clinical complications, including CSF leaks and cranial nerve palsies. Given that some of these complications may be delayed, patient education is critical.

## 1. Introduction

Pediatric head trauma is one of the leading causes of disability and death in children and can be caused by motor vehicle accidents, blunt trauma, falls from heights, and child abuse [1,2,3,4]. It has been reported that, in the United States, pediatric traumatic brain injuries result in nearly 60,000 hospitalizations and over 500,000 emergency department visits [5]. Left untreated, pediatric head trauma has the potential to cause lasting complications, including cognitive impairment and long-term neurological deficits [6,7,8].

Skull base fractures, also known as basilar skull fractures (BSF), are a type of traumatic head injury that involves a break in one or more bones of the base of the skull. The skull base forms the floor of the cranial cavity and is composed of five bones: the temporal, occipital, sphenoid, frontal, and ethmoid bones. With high-velocity blunt force, the bones of the skull base can break, leading to substantial complications, such as cerebrospinal fluid (CSF) leakage and cranial nerve palsies [9,10]. While the epidemiology of skull base fractures is not well characterized, it is reported that between 4 and 20% of children with head trauma will suffer from such an injury [11].

Given the relative rarity of pediatric skull base fractures, their clinical features and management have not been well elucidated. Here, we sought to perform a comprehensive review of the current literature to establish the risk factors, complications, and management options for pediatric skull base fractures. Our goal was to consolidate findings regarding the pathology and obtain a clearer understanding of how it can be best managed.

## 2. Materials and Methods

This review followed the criteria for reporting systematic literature reviews and meta-analyses as defined by the Preferred Reporting Items for Systematic Review and Meta-Analyses (PRISMA) 2020 statement (Appendix A). An organized literature search was performed on PubMed, Embase, and Google Scholar using the following key words: “pediatric”, “skull fracture”, and “basilar skull fracture”. Selected citations were uploaded into Covidence (Cochrane, London, UK). Relevant studies were also obtained from the reference lists of studies obtained from the literature search. Studies were included if they investigated patients under 18 years of age who experienced trauma to the skull base. Articles not written in English, as well as reviews and/or meta-analyses, were excluded from review. Abstract screening, full-text review, and data extraction were then conducted in accordance with the PRISMA guidelines. Figure 1 illustrates the PRISMA flowchart used to guide the systematic review process.

## 3. Results

A total of 55 studies were obtained from the initial search query. Following abstract screening and full-text review, 21 studies were deemed eligible for data extraction and analysis based on the defined inclusion and exclusion criteria.

Table 1 summarizes the inclusion criteria and demographics of each included study. Approximately 64% of the included patients were male, while 36% of patients were female, as averaged across 12 studies. Four of the 21 studies further stratified their study samples by race, with White and Hispanic patients comprising the majority of the analyzed patients. The median age of patients ranged from 3.2 to 12.8 years, with a mean age of 8.68 years. There was notable variance in the incidence of pediatric basilar skull fractures across the 21 studies, with rates as low as 0.0001% and as high as 7.3%. Grigorian et al. (2019) and Ugalde et al. (2018) noted high incidence rates of basilar skull fractures in patients with evidence of cerebrovascular injury (53.2% and 33.9%, respectively) [12,13].

Table 2 outlines the diagnostic findings and criteria utilized by each study to define pediatric skull base fractures. Notable mechanisms of injury were traffic accidents, falls from pre-defined heights, impact from hard objects, and violence, with fall injuries and traffic accidents being noted as the most common causes across nine of the 21 studies. The fracture classification across the 21 studies was highly variable depending on the granularity of the diagnosis. However, temporal bone fractures were reported as the most common in five studies, at rates of approximately 60%, followed by anterior frontal base fractures.

The most dominant tool for diagnosis was a CT scan, as reported by 11 of the 21 studies. Four studies supplemented CT scans with plain film and physical exam findings for a more holistic diagnosis. Common physical exam findings reported across the 21 studies included hemotympanum, Battle’s sign, and raccoon eye. Other reported signs were tympanic membrane rupture, epistaxis, altered mental status, loss of consciousness, and scalp hematoma. Eight of the 21 studies noted intracranial hematoma/hemorrhage as the most common concomitant injury observed in pediatric patients with basilar skull fractures, including epidural, subdural, and subarachnoid hemorrhage types. Other observed concomitant injuries included brain contusion, pneumocephalus, and edema.

Table 3 highlights the reported complications associated with pediatric basilar skull fractures, as well as their management and prognosis. Cranial nerve damage was reported as a common complication, with facial nerve palsy reported in seven of the 21 studies. Of the seven studies, four reported the delayed onset of facial nerve palsy following hospital discharge, with full resolution occurring within three months for most cases. Other observed cranial nerve palsies included olfactory, optic, oculomotor, abducens, glossopharyngeal, and vagus. Moreover, 10 of the 21 studies reported findings of a CSF leak via otorrhea or rhinorrhea, while four studies reported the incidence of in-patient or delayed meningitis. The CSF leak rates were approximately 25%, while the meningitis rates were substantially lower at around 1%. Other noted complications included blunt cerebrovascular injury, hearing loss, eye injuries, central diabetes insipidus, and permanent neurological deficits.

Management was divided into conservative and surgical treatments. Conservative options included IV fluids and antiemetics to stabilize nausea and vomiting, acetazolamide to reduce the intracranial pressure, corticosteroids to treat cranial nerve palsies, and prophylactic antibiotics against meningitis. Surgical treatment was reported in eight of the 21 studies to address the fracture directly or treat complications such as CSF leaks or cranial nerve compression. The average hospital length of stay ranged from 5 to 17 days, with the majority of patients returning to baseline function after 2–3 months. However, approximately 1–5% of patients experienced poor outcomes such as transfer to chronic care facilities and/or a comatose state. Death rates were marginal at approximately 1%, usually in the first week following injury due to massive intracranial hemorrhage.

## 4. Discussion

Pediatric head trauma is one of the leading causes of disability and death in children and can involve underlying trauma to the skull. While pediatric head trauma and its consequences have been well characterized in the literature, there is a noticeable paucity of information regarding pediatric skull base fractures. Furthermore, there is a high degree of variability in the injury characteristics, complications, and management among the few studies that have examined the topic. As such, this review provides a concise and consolidated summary of pediatric basilar skull fractures to guide management for providers and caretakers.

We found that the incidence of pediatric basilar skull fractures ranged from 0.0001 to 7.3%, with the most common mechanisms of injury being traffic-related accidents and fall injuries. In patients with evidence of cerebrovascular injury, however, the incidence rates were significantly higher (between 33.9% and 53.2%). This could be explained by the anatomical location of the skull base relative to the major cerebrovascular blood supply routes within the cranial cavity. In regard to demographics, male patients had a higher incidence of fracture compared to female patients. While this finding may be attributable to the samples of each study, the relationship between sex and the basilar fracture incidence in children may be a worthwhile investigation.

The fracture classification across the 21 studies was highly variable, but the majority of studies reported temporal bone fractures as the most common fracture pattern in pediatric basilar skull injuries, followed by anterior frontal base fractures. This contrasts with adult basilar skull fractures, where anterior frontal base fractures are reported to be the most common. This difference in presentation in the pediatric population may be explained by anatomical differences in the skull shape and structure. As posited by Leibu et al. (2017), the ratio of the skull to facial volume is 1:8 in children and decreases to 1:2.5 by adulthood, thereby increasing the odds of anterior frontal base fractures and decreasing the odds of temporal bone fractures [20]. Furthermore, the maxillary and frontal sinuses are still in development in children, rendering them particularly susceptible to fracture if subjected to blunt trauma.

While most of the studies included in our analysis reported CT as the primary diagnosis tool, there was notable debate about the efficacy of CT scans versus physical exam findings such as Battle’s sign and raccoon eyes. Yildirim et al. (2005) found that only 26% of the patients in their cohort with positive CT findings of a basilar skull fracture displayed identifiable physical exam findings [31]. They argue that Battle’s sign and raccoon eyes were neither sensitive nor specific to basilar skull fractures, thus making CT the ideal tool for diagnosis. In contrast, Tunik et al. (2016) and Kadish and Schunk (1995) emphasized the importance of physical examination [18,30]. They found that only 50% of pediatric patients with basilar skull fractures had positive findings on the CT scan; thus, a strong physical exam can identify patients who have negative findings on imaging. Regardless of their sensitivity in detecting basilar skull fractures, CT scans still are still vital in pediatric patients presenting with signs and symptoms due to the likelihood of associated intracranial injuries that may not be detected by physical examination alone.

A notable correlation between basilar skull fractures and concomitant intracranial hematoma/hemorrhage was noted in our analysis. Though these findings depend on the severity of the injury, the anatomical proximity of major cerebrovascular routes to the skull base likely predisposes patients to epidural, subdural, or subarachnoid hemorrhages in the event of a skull base fracture. Given that pediatric patients with brain bleeds such as these are at a heightened risk of rapid decompensation and death, CT or MRI is recommended for any child presenting with signs of skull fracture, regardless of the location [27].

CSF leakage via otorrhea or rhinorrhea was a reported complication in 10 of the 21 studies in our analysis. Leibu et al. (2017) reported the usage of continuous drainage for management, with reoperation if necessary for repair [20]. Four studies reported the onset of meningitis as another complication, with all suggesting the infection as a sequela of CSF leakage [20,22,25,27]. Given the high rates of meningitis in adult patients with basilar skull fractures, they recommend administering prophylactic antibiotics to pediatric patients to protect against meningeal manifestations. However, given the low incidence rate of around 1% of meningitis infections that we observed in our review, we question the need for prophylactic antibiotics in pediatric patients presenting with skull base fractures.

Cranial nerve palsies were another commonly reported complication of pediatric skull base fractures, with the facial nerve being the most commonly involved nerve. Injury to the facial nerve is likely due to direct damage to the petrous temporal bone, where the nerve commonly tracks. Of note, four studies reported the delayed onset of facial nerve palsy following discharge [20,22,29,31]. As a result, patients should be educated on the signs and symptoms of facial nerve palsy to ensure prompt follow-up if symptoms manifest. Additionally, a syndrome known as Jugular Foramen syndrome was reported by Toledo-Gotor et al. (2021) and Yildirim et al. (2005) [29,31]. Jugular Foramen syndrome is the paralysis of cranial nerves IX–XI as they pass through the jugular foramen, manifesting as dysphagia, hoarseness, and dizziness. While the syndrome is not extensively discussed in the literature, it is a noteworthy complication that providers should consider when providing discharge counseling for patients and their families.

The treatment of pediatric basilar skull fractures included the conservative medical management of associated symptoms and/or surgical intervention to correct deformities or manage bleeding or leakage. Dunnick et al. (2019) reported acetazolamide usage to manage elevated intracranial pressure [16]. Despite being uncommonly used in pediatric patients, acetazolamide was used due to evidence of papilledema as well as temporary therapeutic relief seen with lumbar puncture. Treatment was started in the hospital and continued after discharge, with symptom resolution reported after 60 days. Liu-Shindo and Hawkins (1989) reported corticosteroid usage to manage cranial nerve palsies; however, corticosteroids in patients with TBI have been associated with negative effects such as the exacerbation of critical illness-related corticosteroid insufficiency [22,32]. However, these effects were often seen with high-dose corticosteroid administration and during the first 24–48 h of TBI. Therefore, in pediatric patients who may present with delayed cranial nerve palsies following discharge, corticosteroid treatment remains a viable treatment option.

Surgical intervention was often indicated for patients with intracranial hemorrhage, a CSF leak, or cranial nerve compression. Craniotomies were performed in patients with elevated intracranial pressure to elevate the depressed fracture fragments compressing the cranium. Leibu et al. (2017) reported surgical drain placement as a treatment for patients with CSF leaks via rhinorrhea [20]. Liu-Shindo and Hawkins (1989) performed facial nerve decompression in several patients to relieve cranial nerve palsies from fracture fragments, as well as frontal sinus obliteration and anterior fossa dura repair to treat CSF rhinorrhea [22].

The present study was not without limitations. First, there was inherent variability across the 21 studies in regard to the study design, sample patient population, and outcome measurements. The reported outcomes varied from paper to paper, as some studies did not include variables such as the fracture classification, management, or prognosis. Lastly, there were limitations attached to the study’s search strategy; some relevant studies could have been excluded due to the omission of certain terms in our preliminary search.

Overall, the studies in this review outline the etiology, manifestations, and complications associated with pediatric skull base fractures. Despite their relative rarity, pediatric skull base fractures are associated with significant clinical complications, including CSF leaks and cranial nerve palsies. Our clinical recommendations regarding management include a thorough physical examination alongside rapid imaging to confirm the diagnosis, as well as patient education on delayed cranial nerve palsies that can manifest after discharge.

## Figures and Tables

**Figure 1 children-11-00564-f001:**
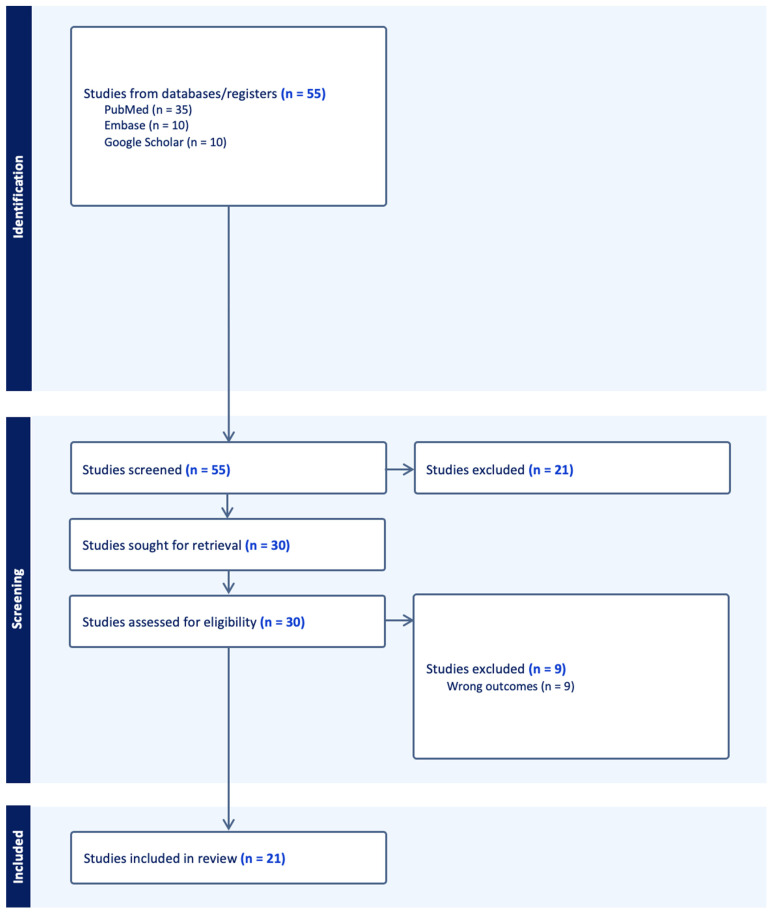
PRISMA flow diagram of the systematic literature search performed (*n* = number of studies).

**Table 1 children-11-00564-t001:** Study characteristics.

Author	Study Design	Indications for Study Inclusion	Demographic Information of Pediatric Patients with BSF	Mean/Median Age of Pediatric Patients with BSF	Incidence of Pediatric Basilar Skull Fractures
Astafyeva et al., 2022 [11]	Case report	N/A	N/A	N/A	N/A
Barba et al., 2022 [14]	Case–control study	Pediatric patients presenting with a multi-level fall	Male: 109/180 (60.5%) Female: 71/180 (39.4%)	Median age: 3.62 years	After multi-level fall: 180/4315 (4.2%)
Bressan et al., 2021 [15]	Case–control study	Pediatric patients presenting to the ED	N/A	N/A	N/A
Dunnick et al., 2019 [16]	Case report	N/A	N/A	N/A	N/A
Garcia et al., 2005 [17]	Case–control study	Pediatric patients with ocular and adnexal injuries	Male: 4573/7065 (64.0%) Female: 2492/7065 (36.0%)	Mean age: 7.45 years	7065/96,879 (7.3%)
Grigorian et al., 2019 [12]	Case–control study	Pediatric patients involved in a blunt trauma mechanism	N/A	N/A	Amongst pediatric trauma patients: Skull base fracture and no BCVI: 4891/60,040 (7.1%) Skull base fracture and BCVI: 58/109 (53.2%)
Kadish et al., 1995 [18]	Case–control study	Pediatric patients presenting to ED with a basilar skull fracture	Male: 147/239 (62%) Female: 92/239 (38%)	Median age: 7 years	N/A
Kopelman et al., 2011 [19]	Case–control study	Pediatric blunt trauma patients	N/A	N/A	N/A
Leibu et al., 2017 [20]	Case–control study	Pediatric patients with skull base fracture	Male: 153/196 (78.1%) Female: 43/196 (21.9%)	Median age: 5 years Mean age: 6.4 years	N/A
Leraas et al., 2019 [21]	Case–control study	Blunt trauma patients	N/A	N/A	N/A
Liu-Shindo and Hawkins, 1989 [22]	Case–control study	Pediatric patients with skull base fracture	Male: 43/62 (69.4%) Female: 19/62 (30.6%)	N/A	N/A
Magit et al., 2021 [23]	Case–control study	Pediatric patients with skull base fracture	Male: 478/729 (65.6%) Female: 251/729 (34.4%) Non-Hispanic White: 32.2% Hispanic/Latino: 45.3% Asian: 3.7% African American: 0.7% Native American: 0/% Native Hawaiin/Pacific Islander: 0.5% Other/unreported: 13.3%	Mean age: 6.78 years	N/A
Mallicote et al., 2019 [24]	Case–control study	Pediatric blunt trauma patients	N/A	N/A	N/A
McCutcheon et al., 2013 [25]	Case–control study	Patients with isolated basilar skull fracture	Male: 2486/3563 (69.8%) Female: 1077/3563 (30.2%) White: 1325/3563 (37.9%) Hispanic: 1550/3563 (44.3%) Black: 228/3563 (6.5%) Asian or Pacific Islander: 181/3563 (5.2%) Native American/other: 214/3563 (6.1%)	Mean age: 8 years Median age: 8 years	N/A
Perheentupa et al., 2012 [26]	Case–control study	Pediatric patients with fracture of the frontal skull base	Male: 11/20 (55%) Female: 9/20 (45%)	Mean age: 12.8 years	0.0006–0.002%
Perheentupa et al., 2010 [27]	Case–control study	Pediatric patients with skull base fracture	Male: 37/63 (58.7%) Female: 26/63 (41.3%)	Mean age: 10.7 years	0.0001–0.0013%
Ryan et al., 2024 [28]	Case–control study	Pediatric trauma patients with simple basilar skull fracture	Male: 107/174 (61.5%) Female: 67/174 (14.1%) African American: 1/174 (0.6%) Asian: 2/174 (1.2%) White: 146/174 (83.9%) Hispanic: 6/174 (3.4%) Other: 19/174 (10.9%)	Median: 3.2 years	N/A
Toledo-Goto et al., 2021 [29]	Case report	N/A	N/A	N/A	N/A
Tunik et al., 2016 [30]	Case–control study	Pediatric patients with blunt head trauma	Male: 330/525 (62.9%) Female: 195/525 (37.1%)	N/A	558/42,958 (1.3%)
Ugalde et al., 2018 [13]	Case–control study	Pediatric trauma patients who underwent computed tomography angiography (CTA)	Male: 246/375 (65.6%) Female: 129/375 (39.4%) White: 193/375 (51.6%) Non-White: 181/375 (48.4%)	N/A	Amongst pediatric trauma patients who underwent CTA imaging: 127/375 (33.9%)
Yildirim et al., 2005 [31]	Case report	N/A	N/A	N/A	N/A

**Table 2 children-11-00564-t002:** Diagnosis and classification of pediatric BSFs.

Author	Mechanism of Injury of BSF	Fracture Classification	Diagnostic Method of Evaluation	Physical Exam Findings	Concomitant Intracranial Injury
Astafyeva et al., 2022 [11]	Traffic accident	Skull base fracture passing through the sella turcica	CT	Initial GCS: 6	Acute severe TBI, subarachnoid hemorrhage
Barba et al., 2022 [14]	Fall < 6 feet: 104/80 (57.8%); Fall > 6 feet and <15 feet: 29/180 (16.1%); Fall > 15 feet: 47/180 (26.1%)	Temporal bone fracture: 109/180 (60.6%); Non-temporal bone fracture: 71/180 (39.4%)	CT	Hemotympanum: 72/180 (40%); Battle’s sign: 6/180 (3.3%); Raccoon sign: 42/180 (23.3%) Initial GCS: 14–15 (83%)	Intracranial bleed: 65/180 (36.1%)
Bressan et al., 2021 [15]	N/A	N/A	N/A	Battle’s sign, racoon eyes, CSF otorrhea, CSF rhinorrhea Initial GCS: 14–15	Clinically important TBI
Dunnick et al., 2019 [16]	Fall from height (3.5 feet)	Right occipital fracture	CT	Grade 2 right optic nerve papilledema Initial GCS: 14	Increased intracranial pressure
Garcia et al., 2005 [17]	N/A	N/A	N/A	N/A	N/A
Grigorian et al., 2019 [12]	N/A	N/A	N/A	Median GCS: 9	N/A
Kadish et al., 1995 [18]	Fall < 5 feet: 47/239 (20%); Fall > 5 feet: 49/239 (20%); Motor vehicle accident: 47/239 (20%); Pedestrian versus vehicle: 34/239 (14%); Bicycle: 27/239 (11%)	N/A	CT: 51/239 (21%); PE: 94/239 (29.3%); CT + PE: 94/239 (39.3%)	Hemotympanum: 122/239 (51%); Battle’s sign: 26/239 (10.9%) Initial GCS: 15 (60%), <15 (40%)	Contusions, Subarachnoid hemorrhages, Intracerebral hemorrhages, Pneumocephalus, Epidural hemorrhages, Subdural hemorrhages
Kopelman et al., 2011 [19]	N/A	N/A	N/A	Initial GCS: ≤8 (31%)	N/A
Leibu et al., 2017 [20]	Fall from height: 143/196 (73%); Motor vehicle accident: 34/196 (17%); Other (falling heavy object, physical violence): 19/196, (10%)	Temporal bone fracture: 112/196 (57%); Frontal base (anterior): 62/196 (32%); Occiput (posterior): 13/196, (7%)	N/A	Initial GCS: 14–15 (87%)	None: 86/196 (43.9%); Bleeding (e.g., EDH, SDH): 60/196 (30.6%); Pneumocephalus: 37/196 (18.9%); Edema: 1/196 (0.5%); Mixed: 32/196 (16.3%)
Leraas et al., 2019 [21]	N/A	N/A	N/A	Mean GCS: 6	N/A
Liu-Shindo and Hawkins, 1989 [22]	Struck by vehicle: 26/62 (40%); Falls: 17/62 (27%); Vehicle accident: 14/62 (23%)	Temporal bone fracture: 30/57 (52.6%); Occipital bone fracture: 4/57 (7.0%); No fracture on scans: 14/57 (24.6%)	CT: 15/57 (26.3%); Plain Film: 22/57 (38.6%); CT + Plain Film: 20/57 (35.1%)	Hemotympanum: 36/62 (58.1%); Blood in ear: 29/62 (46.8%); Hearing loss: 21/62 (33.9%); Tympanic membrane perforation: 16/62 (25.8%); Vestibular symptoms: 5/62 (8.1%); Battle’s sign: 0/62 (0%)	Intracranial hemorrhage: 8/62 (12.9%)
Magit et al., 2021 [23]	Multilevel fall: 261/729 (35.8%); Unhelmeted rider: 136/729 (18.7%); Pedestrian versus vehicle: 56/729 (7.7%); Motor vehicle accident restrained: 48/729 (6.6%); Single-level fall: 46/729 (6.3%); Sport injury: 42/729 (5.8%)	N/A	N/A	N/A	N/A
Mallicote et al., 2019 [24]	N/A	N/A	N/A	Initial GCS: ≤6	N/A
McCutcheon et al., 2013 [25]	N/A	N/A	N/A	N/A	No intracranial injury: 1744/3563 (49%); Laceration or contusion: 356/3563 (10.0%); Hematoma: 1110/3563 (31.2%); Unspecified hematoma: 353/3563 (9.9%)
Perheentupa et al., 2012 [26]	Road traffic accident: 9/20, (45%); Hit by heavy object: 4/20 (20%); Violence: 3/20 (15%); Fall from height: 2/20 (10%); Fall to the ground: 2/20 (10%)	Anterior skull base fracture: 15/20 (75%); Orbital roof fracture: 8/20 (40%); Fracture of posterior wall of the frontal sinus: 4/20 (45%); Sphenoid sinus fracture: 8/20 (40%); Injury of cribriform plate: 8/20 (40%); Injury of middle part of skull base: 5/20 (25%)	CT	Altered consciousness or unconscious: 15/20 (75%) Mean GCS: 10	Brain contusion: 12/20 (60%); Pneumocephalus: 11/20 (55%); Subarachnoid bleeding: 10/20 (50%); Intracranial hematoma: 6/20 (30%); Subdural bleeding: 4/20 (20%); Intracranial edema: 3/20 (15%)
Perheentupa et al., 2010 [27]	Road traffic accident: 24/63 (38.1%); Falling from height: 20/63 (31.7%); Falling to the ground: 6/63 (9.5%); Violence: 5/63 (7.9%)	Temporal bone fracture: 40/63 (63.5%); Sphenoethmoidal complex injury: 26/63 (41.3%); Orbit injury: 22/63 (34.9%); Occipital bone fracture: 10/63 (15.9%); Parietal bone fracture: 7/63 (11.1%); Frontobasal fracture: 14/63 (22.2%)	CT: 57/63 (90.5%); PE: 6/63 (9.5%); X-ray: 2/63 (3.2%)	Unconscious when primarily met: 22/63 (34.9%); Altered consciousness or intubated and consequently sedated: 36/63 (57.1%); Hemotympanum: 39/63 (61.9%); Hearing loss: 30/63 (47.6%); External auditory canal bleeding: 29/63 (46.0%); Tympanic membrane perforation: 6/63 (9.5%); Epistaxis: 13/63 (20.6%); Raccoon Eye: 21/63 (33.3%); Vertigo: 10/63 (15.9%) Mean GCS: 13	Brain contusion/hematoma: 15/27 (55.6%); Subdural hematoma: 8/27 (29.6%); Edema and cerebral bleeding: 2/27 (7.4%); Epidural hematoma: 1/27 (3.7%); Subarachnoid hematoma: 1/27 (3.7%)
Ryan et al., 2024 [28]	N/A	Anterior fossa fracture: 38/174 (21.8%); Middle fossa fracture: 57/174 (32.8%); Posterior fossa fracture: 79/174 (45.4%)	CT	Initial GCS: 15	Delayed intracranial hemorrhage: 0/174 (0%)
Toledo-Goto et al., 2021 [29]	Hit by a motor vehicle	Bilateral skull base fracture (involving left petrous bone)	CT	Right temporoparietal hematoma of the scalp, contusion with incisive wound on left ear, bilateral blood otorrhea, epistaxis Initial GCS: 11	N/A
Tunik et al., 2016 [30]	Fall from elevation: 123/525 (23.4%); Occupant in motor vehicle crash: 90/525 (17.1%); Pedestrian struck by moving vehicle: 59/525 (11.2%); Assault: 37/525 (7.0%); Object struck head, accidental: 34/525 (6.5%); Other wheeled transport crash: 30/525 (5.7%); Bike rider struck by automobile: 25/525 (4.8%); Bike crash or fall from bike while riding: 18/525 (3.4%)	N/A	CT: 162/525 (30.9%); PE: 292/558 (52.3%); CT + PE: 104/525 (19.8%)	Hemotympanum: 203/363 (55.9%); Racoon eye: 82/363 (22.6%); Battle’s sign: 35/363 (9.6%) Initial GCS: 15 (60%), <13 (32%)	Pneumocephalus: 145/525 (27.6%); Subarachnoid hemorrhage: 70/525 (13.3%); Cerebral hemorrhage/intracerebral hematoma: 67/525 (12.8%); Subdural hematoma: 66/525 (12.5%); Cerebral contusion: 64/525 (12.2%); Cerebral edema: 48/525 (9.1%)
Ugalde et al., 2018 [13]	N/A	N/A	N/A	Initial GCS: >8 (66%), ≤8 (33%)	N/A
Yildirim et al., 2005 [31]	Fall from roof of house	Temporal bone fracture (involving tip of petrous pyramid)	CT	Otorrhea with hemorrhage in the left ear, hematoma of scalp on the left temporoparietal location Initial GCS: 9	N/A

**Table 3 children-11-00564-t003:** Complications and management of pediatric BSFs.

Author	Incidence and Duration of Cranial Nerve Injury	Incidence and Duration of CSF Leakage and Meningitis	Other Complications	Management	Prognosis/Survivability
Astafyeva et al., 2022 [11]	N/A	CSF leak present	Central diabetes insipidus, amaurosis, left-sided hemiparesis	Oral therapy with desmopressin, elevation of depressed fracture	N/A
Barba et al., 2022 [14]	Facial nerve injury: 19/180 (10.6%)	CSF leak: 7/180 (3.9%)	Hearing loss: 22/180 (12.2%); Vertigo: 3/180 (1.7%)	N/A	N/A
Bressan et al., 2021 [15]	N/A	N/A	N/A	Craniotomy, elevation of depressed fracture	N/A
Dunnick et al., 2019 [16]	N/A	N/A	Vomiting	Anti-emetics, acetazolamide 10 mg/kg orally twice a day	Return to baseline after 2 months
Garcia et al., 2005 [17]	N/A	N/A	Eye injury	N/A	N/A
Grigorian et al., 2019 [12]	N/A	N/A	Blunt cerebrovascular injury	N/A	N/A
Kadish et al., 1995 [18]	N/A	CSF leak: 46/239 (19%)	N/A	N/A	N/A
Kopelman et al., 2011 [19]	N/A	N/A	Blunt cerebrovascular injury	N/A	N/A
Leibu et al., 2017 [20]	Facial nerve palsy and oculomotor palsy present On discharge: 8/196 (4%) On follow-up: 11/124 (9%)	CSF leak: 54/196 (27.6%); Meningitis: 2/54 patients with CSF leak (3.7%)	Vomiting: 36/196 (18.4%); Loss of consciousness: 27/196 (13.8%); Hearing loss: on discharge, 8/196 (4%); on follow-up, 20/196 (16%)	Treatment for CSF leak: Conservative (47/54, 88.9%) Continuous drainage (3/54, 5.6%) Continuous drainage + operation (4/54, 5.6%)	Mean duration of hospitalization: 5.4 days Majority of patients recovered uneventfully
Leraas et al., 2019 [21]	N/A	N/A	Carotid artery injury	N/A	N/A
Liu-Shindo and Hawkins, 1989 [22]	Facial paralysis: 8/62 (12.9%); delayed onset in 5 cases 6 completely resolved in less than 3 months	CSF otorrhea: 16/62 (25.8%) CSF rhinorrhea: 1/62 (1.6%) Meningitis: 2/62 (3.2%)	Coma: 3/62 (4.8%)	4/62 (6.5%): treated with antibiotics; 8/62 (12.95%): corticosteroids; 1/61 (1.6%): surgical decompression of the facial nerve; 1/61 (1.6%): frontal sinus obliteration and repair of anterior fossa dura	1/62 (1.6%): died due to massive intracranial hemorrhage 3/62 (4.8%): transferred to chronic care facilities in a comatose state
Magit et al., 2021 [23]	N/A	N/A	N/A	N/A	N/A
Mallicote et al., 2019 [24]	N/A	N/A	Blunt cerebrovascular injury	N/A	N/A
McCutcheon et al., 2013 [25]	N/A	CSF leak: 2.33% Meningitis among patients without CSF leak: 0.4% Meningitis among patients with CSF leak not requiring repair: 1.9% Meningitis among patients with CSF leak requiring repair: 4.3%	N/A	N/A	N/A
Perheentupa et al., 2012 [26]	Olfactory nerve dysfunction: 2/20 (10%)	CSF leak: 5/20 (25%)	Opthalmic problems (ptosis, diplopia, telecanthus, enophthalmos): 8/20 (40%) Permanent neurological problems: 5/20 (25%)	12/20 (50%): treated surgically 13/20 (65%): treated in the ICU	Mean length of hospital stay: 17 days; 16/20 (80%): discharged home; 3/20 (15%): transferred to another institution; 1/20 (5%): died after 1 week of intensive care treatment
Perheentupa et al., 2010 [27]	Facial nerve palsy: 3/63 (4.8%), 1 permanent Optic nerve deficiency: 4/63 (6.3%), all permanent Abducens nerve deficiency: 3/63 (4.8%), all temporary Olfactory nerve deficiency: 2/63 (3.2%), all permanent	CSF leak: 7/63 (11.1%) Meningitis: 1/63 (1.6%)	Permanent neurological deficits: 10/63 (15.9%)	16/63 (25.4%): treated operatively 47/63 (74.6%): treated conservatively 40/47 (81.1%) had prophylactic antibiotic therapy	Mean hospital stay: 10.4 days 1 patient (1.5%) died after 1 week 54/63 (85.7%) were discharged home 8/63 (12.7%) were discharged to another institution for inpatient rehabilitation
Ryan et al., 2024 [28]	Facial nerve injury: 2/174 (1.2%)	CSF leak: 3/174 (1.7%) Meningitis: 0/174 (0%)	Otitis media: 2/174 (1.2%); Severe nausea/vomiting: 10/174 (5.7%)	10/174 (5.7%) received IV fluids due to vomiting/dehydration	Deaths: 0/174 (0%); Hospitalization for >24 h: 30/174 (17.2%); Return to hospital within 3 weeks of original injury: 8/174 (4.5%)
Toledo-Goto et al., 2021 [29]	Delayed lower cranial nerve palsy; glossopharyngeal nerve palsy; vagus nerve palsy; facial nerve palsy (all resolved by 3 months post-admission)	N/A	Hearing loss	Steroids and physical therapy	Near full recovery by 3 months
Tunik et al., 2016 [30]	Cranial nerve involvement: 22/457 (4.8%)	CSF otorrhea: 32/363 (8.9%)CSF rhinorrhea: 18/363 (5.0%)	N/A	22% (72/324) Neurosurgery	5.7% (32/558) of patients died; 76/558 (13.7%) patients were intubated for at least 24 h
Ugalde et al., 2018 [13]	N/A	N/A	Blunt cerebral vascular injury	Aspirin, anticoagulation, surgery/invasive intervention	N/A
Yildirim et al., 2005 [31]	Delayed lower cranial nerve palsy: glossopharyngeal nerve palsy; facial nerve palsy; vagus nerve palsy (all improved after a month)	N/A	Sensorineural hearing loss	Narcotics and benzodiazepines for initial treatment Steroids after onset of cranial nerve palsies	Most of the cranial nerve palsies resolved after a month

## Data Availability

No new data were created or analyzed in this study. Data sharing is not applicable to this study.

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
