# Peer review of "Clinical Features and Management of Skull Base Fractures in the Pediatric Population: A Systematic Review"

_children, 2024, doi:10.3390/children11050564_

Round 1

Reviewer 1 Report

Comments and Suggestions for Authors

I appreciate the authors for presenting this compressive review article emphasizing the etiology, diagnosis, clinical pictures and complication for pediatric skull base fracture. The authors also provide clinical recommendations regarding management and diagnosis. This is a well-reviewed and organized review article. The major drawback is the heterogeneous of the included 21 studies. However, the summarized information may provide to the readers interested in the field of pediatric trauma.  My comments are as followings

1. Abstract: the authors mentioned the “Pediatric BSFs often presented with intracranial hematoma/hemorrhage (8/21),….”. This result was based on the 21 study. It seemed not proper. It would be better to present with a range of incidence   in different study.

2. In Table 2, it showed concomitant Intra-cranial Injury aroud 0-55 %. It would be clear if the authors provide the initial GCS in Table 2.

Author Response

Reviewer 1:

I appreciate the authors for presenting this compressive review article emphasizing the etiology, diagnosis, clinical pictures and complication for pediatric skull base fracture. The authors also provide clinical recommendations regarding management and diagnosis. This is a well-reviewed and organized review article. The major drawback is the heterogeneous of the included 21 studies. However, the summarized information may provide to the readers interested in the field of pediatric trauma.  My comments are as followings

  1. Abstract: the authors mentioned the “Pediatric BSFs often presented with intracranial hematoma/hemorrhage (8/21),….”. This result was based on the 21 study. It seemed not proper. It would be better to present with a range of incidence   in different study. 

 - We agree with the reviewer’s assessment here and have modified Lines 17-18 accordingly to report incidence rate of intracranial hematoma/hemorrhage across the 21 studies.

  1. In Table 2, it showed concomitant Intra-cranial Injury aroud 0-55 %. It would be clear if the authors provide the initial GCS in Table 2.

- We agree with the reviewer’s recommendation here and have included either initial or mean GCS scores as reported by the 21 studies in Table 2.

Reviewer 2 Report

Comments and Suggestions for Authors

This study systematically reviewed the available literature regarding pediatric BSF after TBI. However, I had several concerns regarding this study. 

1. The authors only used one database in conducting this review (PubMed). A minimum of three databases is needed to conduct a systematic review. 

2.  Cranial nerve palsy is a common complication of BSF. The authors mentioned it and stated the use of corticosteroids to manage it. However, the corticosteroid is correlated with many negative effects if given in the acute period of TBI. Please elaborate more about this in the discussion section. 

3. Acetazolamide is not commonly used to manage increased ICP in pediatrics. Please elaborate more about it in the discussion section as well. 

4. Surgical treatment was conducted in eight studies to manage the skull fracture associated with CSF leakage. Discussing more about this finding would make this paper more interesting.

Congratulation for the work. 

Author Response

Reviewer 2:

This study systematically reviewed the available literature regarding pediatric BSF after TBI. However, I had several concerns regarding this study. 

  1. The authors only used one database in conducting this review (PubMed). A minimum of three databases is needed to conduct a systematic review.

- PubMed, Embase, and Google Scholar were utilized to collate the 21 studies analyzed in this review. We found no specific mentions of the databases we used in our initial draft, so we have added them in for further clarification (Lines 49-51). We thank the reviewer for highlighting this issue.

  1.  Cranial nerve palsy is a common complication of BSF. The authors mentioned it and stated the use of corticosteroids to manage it. However, the corticosteroid is correlated with many negative effects if given in the acute period of TBI. Please elaborate more about this in the discussion section.

- We agree with the reviewer’s suggestion here. Lines 190-196 highlight the negative effects associated with corticosteroid use in patients with TBI, as well as justification for why they can be safely used in patients who present with delayed cranial nerve palsies following BSF

  1. Acetazolamide is not commonly used to manage increased ICP in pediatrics. Please elaborate more about it in the discussion section as well. 

- We agree with the reviewer’s suggestion here. Lines 184-190 describe the clinical indications for acetazolamide as well as the treatment outcomes for the patient in which it was utilized.

  1. Surgical treatment was conducted in eight studies to manage the skull fracture associated with CSF leakage. Discussing more about this finding would make this paper more interesting.

- We agree with the reviewer’s suggestion here. Lines 197-203 highlight the different surgeries performed for pediatric patients as well as their indications.

If there are any more questions or concerns, please do not hesitate to reach out and we will be more than happy to help. Thank you again for considering of our manuscript for publication.

Round 2

Reviewer 1 Report

Comments and Suggestions for Authors

The revised manuscript has replied my comments item-by-item. I have nor more comments. 

Author Response

We thank the reviewer for their comments.

Reviewer 2 Report

Comments and Suggestions for Authors

Thank you for addressing all of my comments. There is an improvement of the overall manuscript. However, there was only one database mentioned on figure 1. Please make change to it. 

Congratulation. 

Author Response

We thank the reviewer for the comment-- Figure 1 has been updated.